

# Genome-wide identification of new reference genes for RT-qPCR normalization in CGMMV-infected *Lagenaria siceraria*

Chenhua Zhang[1,2,*], Hongying Zheng[2,3,*], Xinyang Wu[2,3], Heng Xu[3], Kelei Han[2,3], Jiejun Peng[2,3], Yuwen Lu[2,3], Lin Lin[2,3], Pei Xu[2,4], Xiaohua Wu[2,4], Guojing Li[2,4], Jianping Chen[1,2,3] and Fei Yan[2,3]

[1] College of Life Sciences, Fujian Agriculture and Forestry University, Fuzhou, China
[2] The State Key Laboratory Breeding Base for Sustainable Control of Pest and Disease, Zhejiang Academy of Agricultural Sciences, Hangzhou, China
[3] Key Laboratory of Biotechnology in Plant Protection of MOA of China and Zhejiang Province, Institute of Virology and Biotechnology, Zhejiang Academy of Agricultural Sciences, Hangzhou, China
[4] Institute of Vegetable, Zhejiang Academy of Agricultural Sciences, Hangzhou, China
[*] These authors contributed equally to this work.

Corresponding authors
Hongying Zheng, zhenghongyinghz@163.com
Fei Yan, fei.yan@mail.zaas.ac.cn

## ABSTRACT

*Lagenaria siceraria* is an economically important cucurbitaceous crop, but suitable reference genes (RGs) to use when the plants are infected by cucumber green mottle mosaic virus (CGMMV) have not been determined. Sixteen candidate RGs of both leaf and fruit and 18 candidate RGs mostly from separate RNA-Seq datasets of bottle gourd leaf or fruit were screened and assessed by RT-qPCR. The expression stability of these genes was determined and ranked using geNorm, NormFinder, BestKeeper and RefFinder. Comprehensive analysis resulted in the selection of *LsCYP*, *LsH3*, and *LsTBP* as the optimal RGs for bottle gourd leaves, and *LsP4H*, *LsADP*, and *LsTBP* for fruits. *LsWD*, *LsGAPDH*, and *LsH3* were optimal for use in both leaves and fruits under the infection of CGMMV. Isopentenyl transferase (*IPT*) and DNA-directed RNA polymerase (*DdRP*) were used to validate the applicability of the most stable identified RGs from bottle gourd in response to CGMMV. All the candidate RGs performed in RT-qPCR consistently with the data from the transcriptome database. The results demonstrated that *LsWD*, *LsGAPDH* and *LsH3* were the most suitable internal RGs for the leaf, and *LsH3*, *LsGAPDH*, *LsP4H* and *LsCYP* for the fruit.

## INTRODUCTION

*Lagenaria siceraria* (Molina) Standl. is a specie belongs to Cucurbitaceae family, which was cultivated widely in tropical and temperate regions of the world, it is commonly known as bottle gourd that has good edible, medicinal and horticultural value (*Wang et al., 2018*; *Decker-Walters et al., 2004*). It could be routinely used as one rootstock source for watermelon and other cucurbit crops in both Japan and Korea in order to reduce the incidence of soil-borne diseases and promote the vigor of the root system of the crops in low temperature conditions (*Yetisir & Sari, 2013*; *Cho et al., 2017*; *Spalholz & Kubota*,

*2017*). Medicinally, *L. siceraria* extract has radioprotective potential in radiation-induced gastrointestinal injury (*Sharma, Goel & Chauhan, 2016*), and its latex sap exhibits potent lectin activity to mitigate neoplastic malignancy by targeting neovasculature and cell death (*Vigneshwaran et al., 2016*). Recently, a dedicated database named GourdBase was developed, which promoted the study of biological traits and molecular breeding in the bottle gourd (*Wang et al., 2018*). Zhejiang province has a long history of cultivating bottle gourd as an important economic crop. In 2011, the leaves of a bottle gourd plant which were brittle and had severe mosaic mottling were shown to be infected with cucumber green mottle mosaic virus (CGMMV) using reverse transcription-polymerase chain reaction (RT-PCR) and ELISA (*Zheng et al., 2015*). Since CGMMV could pose a great threaten to bottle gourd production, it attracted our attention.

CGMMV (genus *Tobamovirus*, family *Virgaviridae*) causes serious diseases in cucurbit crops. The virus is easily transmitted on the outside of seeds, pollen, and other propagation materials. It produces severe mosaic symptoms on the leaves of infected plants and causes fruit deformation, resulting in reduced yield and low market value (*Ugaki et al., 1991*; *Sano et al., 1997*; *Tan et al., 2000*; *Zheng et al., 2015*; *Ali, Mohammad & Khattab, 2012*). It has a worldwide distribution, and has been reported from many countries, including Israel, China, Greece, USA, Saudi Arabia and Russia (*Ali, Mohammad & Khattab, 2012*; *Zheng et al., 2015*; *Slavokhotova et al., 2007*; *Antignus et al., 1990*; *Varveri, Vassilakos & Bem, 2002*; *Ali, Natsuaki & Okuda, 2004*; *Amer, 2015*).

In view of the significant economic losses to cucurbit crops caused by CGMMV, most research has focused on its detection and control. The interaction between CGMMV and its hosts has gained increasing attention recently, but knowledge about it is still limited. Several studies have focused on identifying novel and conserved microRNAs in response to CGMMV infection or virus-derived siRNAs in a CGMMV infected host and exploring the pathogenic mechanisms from the perspective of protein expression levels in its hosts (*Liu et al., 2015*; *Li et al., 2016*; *Sun, Niu & Fan, 2017*). Internal changes in the host involve the host-virus interaction system, which is often mediated at the transcriptional level, thereby altering gene expression and possibly indirectly affecting plant performance. Quantitative RT-PCR (RT-qPCR) has become the most common method for quantifying and comparing gene expression levels during virus infection because of its rapidity, sensitivity, and specificity (*Radonić et al., 2014*; *Huggett et al., 2005*; *Ceelen, De Craene & De Spiegelaere, 2014*). Reference genes (RGs) are used to minimize experimental errors and normalize the experimental data but these are not universal; different RGs are needed under different experimental conditions. In cucurbits, only a few RGs with different traits have been established, and there are no reports of RGs suitable for use with CGMMV-infected bottle gourd.

Several reference genes have been utilized for reliable RT-qPCR in cucurbit crops, including actin (*ACT*), elongation factor 1 alpha subunit (*EF1α*), glyceraldehyde-3-phosphate (*GAPDH*), serine/threonine-protein phosphatase PP2A catalytic subunit (*PP2A*), Ran-GTPase (*RAN*), 40S ribosomal protein S15-4 (*RPS15*), tubulin alpha (*TUA*), peptidyl-prolyl cis-trans isomerase (*CYP*), 60S ribosomal protein L23 (*RPL23*), ADP-ribosylation factor (*ADP*), ubiquitin-60S ribosomal protein L40 (*UBA*) and transcription

initiation factor TFIID TATA-box-binding protein (*TBP*) (*Kong et al., 2016*; *Kong et al., 2014a*; *Wan et al., 2010*; *Wang et al., 2014*; *Warzybok & Migocka, 2013*; *Sestili et al., 2014*; *Wu et al., 2016*; *Kong et al., 2014b*; *Kong et al., 2015*). In *Nicotiana benthamiana*, *PP2A*, *F-Box* and *L23* are known to be the most stable RGs for exploring plant-virus interactions (*Liu et al., 2012*).

In this study, traditional reference genes were screened as candidate RGs and new, previously unreported, RGs were also sought. Systematic transcriptome analyses, including RNA-Seq and DNA microarray, have been widely used in the study of host-virus interaction recently. Because transcriptome data provide a valuable resource that can be used to determine appropriate RGs (*Kudo et al., 2016*; *Guo, Jiang & Xia, 2016*; *Marcolino-Gomes et al., 2015*; *Zhang et al., 2014*; *Liu et al., 2018*), we screened potential internal RGs from the transcriptome database of bottle gourd infected by CGMMV. We set the corresponding screening parameters to select the candidate genes from the bottle gourd transcriptome database. 11 candidate RGs from leaves and 22 from fruits were selected, including a histon H3 gene (*LsH3*) and a tryptophan and aspartic acid (*WD*)-repeat protein (*LsWD*), which matched the screening parameters for both leaves and fruits.

The stability and suitability of all selected candidate RGs expression was estimated using several algorithms: geNorm, NormFinder, BestKeeper and RefFinder. These algorithms together provide an approach to identify the most stably transcribed new genes (i.e., in addition to the traditional reference genes). Because there is little information on RGs that can be used to normalize gene expression data in CGMMV-infected bottle gourd, we evaluated the selected candidate genes by RT-qPCR, focusing on novel reference gene selection and analysis in CGMMV-infected leaves and fruits. Moreover, parallel analyses on the expression profiles of an Isopentenyl transferase (*IPT*) gene and a DNA-directed RNA polymerase (*DdRP*) gene normalized by the identified RGs were performed to demonstrate the reliability of these identified RGs.

## MATERIALS & METHODS

### Plant preparation, virus inoculation

The cultivation and management of bottle gourd (*L. siceraria*, accession "Hangzhou Gourd") were performed as follow: after soaking and germination, the seeds were first transplanted into 10 cm nutrient preparations with soil rich in organic matter, and when the seedlings grew to two and a half leaf stage, transplanted them into 20 L PVC drums (1 per barrel). The mixed substrate used was peat: vermiculite: perlite: organic fertilizer = 4:4:1:1, The pH of the culture substrate was about 7.0 and the water content was maintained at about 70% relative humidity. A "flower-free" nutrient solution was used once a week (N:P:K = 20:20:20) (Shanghai Yongtong Chemical Co., Ltd., Shanghai, China). The greenhouse conditions were daily temperature 25–28 °C, night temperature 18–20 °C; photoperiod 14 h/d (light intensity is greater than 87.5 $\mu$mol m$^{-2}$ s$^{-1}$). Scaffolding, topping, pruning and pollination were carried out according to routine management. The fruits were harvested 10 days after pollination.

CGMMV inoculum (CGMMV-ZJ) was sap from *L. siceraria* plants with typical symptoms that had been infected with a CGMMV infectious clone 14 d earlier (*Zheng*

*et al., 2015*). At least six plants were inoculated with CGMMV-infected sap at the two and a half leaf stage on the two expanding leaves. Approximately 1 g of plant tissue was homogenized in 20 volumes of inoculation buffer (0.1 M phosphate buffer, pH 7.5, 0.2% sodium sulfite and 0.01 M 2-mercaptoethanol), while the mock plants were only inoculated with buffer.

## RNA sequencing

According to the protocol of TruSeq Small RNA Sample Prep Kits (Illumina, San Diego, CA, USA), the total RNA of about 5 $\mu$g was extracted for the preparation of small RNA library. Sequencing of the RNA-Seq libraries was carried out on an Illumina Hiseq2500 at LC-BIO (Hangzhou, China) following the manufacturer's protocol.

## RNA and first strand cDNA preparation

Three replicate samples of flesh tissue of the ripe fruits and newly expanding leaves from both inoculated and control plants were collected for RNA extraction. Total RNA was extracted from each sample using TRIzol reagent (Invitrogen, Carlsbad, CA, USA) according to the manufacturer's instructions. The RNA quantity and quality from each sample was evaluated by denaturing agarose gel electrophoresis and microfluidic capillary electrophoresis with the Agilent 2100 bioanalyzer (Agilent Technologies, Santa Clara, CA, USA). Only RNA samples with a complete band and A260/A280 ratio in the range 1.8–2.0 were used for the next step. All RNA samples were stored at −70 °C. For virus detection, the first strand cDNA was synthesized using ReverTra Ace-$\alpha$-® kit (TOYOBO, Japan) following the product's protocol. The infection of CGMMV in the tissues was confirmed by CGMMV specific primers, and the primers of ZYMV and WMV were also used to monitor the presence of these two common viruses occurred in cucurbit crops (*Heeju et al., 2015*). For RT-qPCR, first strand cDNA was synthesized from 1 $\mu$g total RNA using PrimeScript$^{TM}$ RT reagent Kit with gDNA Eraser (Perfect Real Time) kit (TaKaRa, Dalian, China) according to the manufacturer's instructions. The negative controls without PrimeScript RT Enzyme Mix I were analyzed in parallel to detect the presence of genomic DNA contamination in the RNA samples.

## Selection of candidate RGs

Partial candidate RGs in leaves and fruits were selected from publicly available references, but most were from our RNA sequencing data. To obtain RGs that are stably and moderately or highly expressed in CGMMV-infected leaves, we kept the Reads Per Kilobases per Million reads (RPKM), ratio of the maximum to the minimum RPKM (RPKMmax/min), and coefficient of variation (CV) to >40, <2.0, and <0.3 at $p < 0.05$, respectively. In fruit, the RPKM, RPKMmax/min, and CV were maintained at >40, <2.0, and <0.2 at $p < 0.05$, respectively. All selected internal RGs have only one transcript and were ranked from small to large according to their RPKMmax/min values.

To select better RGs for both leaves and fruits, the RPKM and RPKMmax/min were kept at >40 and <2.0 for the RGs commonly used in cucurbit plants in keeping with the RNA-seq data. The RGs for leaves and fruits were screened and analyzed simultaneously

with 14 common RGs of cucurbit crops in previous studies. A total of 16 RGs from bottle gourd leaves and fruits were screened and analyzed.

## Primer design and verification of selected gene amplicons

The fourteen common RGs were amplified according to the references or based on primers designed by Primer-BLAST of the RNA sequence data of leaves and fruits (Table 1). Specific primers for the candidate RGs from our RNA-sequencing data were designed using Primer 3 (http://primer3.ut.ee/) (Table 2). All PCR amplicon lengths were between 80–200 bp. All primers were synthesized by a commercial supplier (Biosune, Hangzhou, China).

To check the specificity of all primers, the cDNA of each sample was amplified by PCR, and the amplified products were separated by electrophoresis on 3% agarose gel and purified using a QIAquick Gel Extraction Kit (Qiagen, Hilden, Germany) according to the manufacturer's instructions, and cloned into pEASY-Blunt zero (Transgen, Beijing, China) followed by sequencing.

The quantification cycle (Cq) values obtained by qRT-PCR on a standard curve generated from a fourfold dilution series of one sample at six dilution points for three technical replicates were used to draw the standard curve to get R2 and slope values. The PCR amplification efficiency of each primer was calculated using the equation: $E(Efficiency)\% = (10^{[-1/slope]} - 1) \times 100\%$.

## Quantitative real-time PCR

qRT–PCR was carried out in 384-well plates using the QuantStudio 6 Flex real-time PCR detection system (ABI, USA). Each reaction mixture consisted of 5 µL SYBR Green Realtime PCR Master Mix (TaKaRa, Dalian, China), 0.5 µL cDNA diluted fivefold, 0.5 µL (10 mM) each of forward and reverse primers, and 3.5 µL RNA-free $H_2O$, equating to a final volume of 10 µL in each well. The qPCR reaction was as follows: initial denaturation at 95 °C for 5 min and 40 cycles of amplification (95 °C 15 s, 58 °C 20 s and 72 °C 20 s). Subsequently, fluorescence acquisition was performed after each cycle. A melting curve was generated after 40 cycles of amplification by heating at 65–95 °C. Cq values and baseline were set automatically by the QuantStudio Real-Time PCR Software v1.2 (ABI, USA) using default parameters.

## Gene expression stability analysis

The programs geNorm (*Vandesompele et al., 2002*), NormFinder (*Andersen, Jensen & Orntoft, 2004*), BestKeeper (*Pfaffl et al., 2004*) and RefFinder (http://150.216.56.64/referencegene.php?type=reference) were used to analyze the stability of the candidate RGs under CGMMV infection conditions. All software packages were used according to the manufacturer's instructions.

## Validation of the selected RGs

*LsIPT* and *LsDdRP* genes in CGMMV-infected bottle gourd leaf and fruit tissue were selected to detect the effectiveness of these identified RGs. Primers for the two genes were designed as described above and listed in Table 1. The best RGs identified by the algorithms above were used for normalization.

Zhang et al. (2018), *PeerJ*, DOI 10.7717/peerj.5642

Peer

**Table 1** Primer sequences and PCR amplification characteristics for commonly-used candidate reference genes, *LsIPT* and *LsDdRP*.

| Gene name | Description | NCBI Homolog locus | Gourdbase Homolog locus | Forward primer sequence/reserve primer sequenc (5′–3′) | Product size (bp) | Tm (°C) | Leaf | | Fruit | |
|---|---|---|---|---|---|---|---|---|---|---|
| | | | | | | | *E* (%) | *R²* | *E* (%) | *R²* |
| *LsH3*[a,b] | histone H3 | LOC103494422 | BG_GLEAN_10002693 | CAAACTGCCCGTAAGTCCAC/ GGCTTCTTCACTCCTCCTGT | 101 | 81.9 | 110.06 | 0.9980 | 103.67 | 0.9986 |
| *LsWD*[a,b] | WD repeat-containing protein | LOC101218407 | BG_GLEAN_10012213 | TCTGTGGTACTCGAGAAGGC/ GAGAAATCTCCGGTGTGTCGT | 95 | 82.5 | 106.96 | 0.9987 | 103.13 | 0.9937 |
| *LsACT* | actin-7 | LOC103499652 | BG_GLEAN_10004800 | GGCAGTGGTTGTGAACATGT/ CCCATGCTATCCTCCGTCTT | 98 | 82.1 | 90.11 | 0.9989 | 97.26 | 0.9999 |
| *LsUBA52* | ubiquitin-60S ribosomal protein L40 | LOC101205082 | – | AAGTGTGGACACAGCAACCA/ GGGAAAGAGCCAAAAATAGG | 255 | 79.3 | 94.86 | 0.9990 | 116.23 | 0.9816 |
| *LsRPL23*[a] | 60S ribosomal protein L23 | LOC101203845 | BG_GLEAN_10013074 | CATGACCATATCACCAACACAA/ CGACAATACAGGAGCTAAGAA | 100 | 77.3 | 98.16 | 0.9978 | 97.15 | 0.9942 |
| *LsRAN* | Ran-GTPase | LOC111496316 | BG_GLEAN_10011129 | TCTACTGTTGGGATACCGCT/ CAGAGATCACGATGCCATGTT | 145 | 80.6 | 103.26 | 0.9976 | 90.81 | 0.9963 |
| *LsPP2A*[a,b] | serine/threonine-protein phosphatase PP2A catalytic subunit | LOC103502598 | BG_GLEAN_10010727 | GGCAGATAACTCAAGTTTATGGA/ GCTGTAAGAGGTAAATAATCAAAGAGG | 109 | 75.0 | 92.68 | 0.9993 | 110.62 | 0.9933 |
| *LsGAPDH*[b] | Glyceraldehyde-3-phosphate dehydrogenase | LOC103496285 | – | CCCAGGGGATATCTGCAGGG/ CATGGTGTTTTCAATGGAACCA | 109 | 85.2 | 100.97 | 0.9986 | 91.06 | 0.9933 |
| *LsEF1 α*[a] | Elongation factor 1-α | LOC101215193 | – | CTGCTTGCTCCTGCGTGAAA/ CCACGATGTTGATGTGAATCTTCTC | 118 | 83.9 | 105.42 | 0.9977 | 103.34 | 0.9962 |
| *LsADP*[a,b] | ADP ribosylation factor | LOC101217563 | – | ATATTGCCAACAAGGCGTAGA/ TGCCCGTAAACAATGGACAAA | 92 | 80.2 | 95.04 | 0.9966 | 93.69 | 0.9968 |
| *LsTUA*[b] | tubulin alpha | LOC103502708 | BG_GLEAN_10002510 | AGGACTGGGACGTACCGACA/ CGGCTAATTTTCGCACTCGG | 145 | 83.9 | 103.16 | 0.9901 | 107.20 | 0.9963 |
| *LsTBP*[a,b] | transcription initiation factor TFIID TATA-box-binding protein | LOC103492411 | BG_GLEAN_10001318 | AAACTCTTCCCGCTTCCTCA/ AGCCTTGATCTGCCATTCCT | 143 | 81.3 | 93.79 | 0.9987 | 96.90 | 0.9986 |
| *LsRPS15*[a] | 40S ribosomal protein S15-4 | LOC101217711 | BG_GLEAN_10016309 | AGTCCTCTTCTTCGGCACTC/ TCCACTCGAAACCCTAGCAG | 135 | 80.2 | 90.16 | 0.9988 | 92.14 | 0.9999 |
| *LsCYP20* | peptidyl-prolyl cis-trans isomerase CYP20-1 | LOC101213040 | BG_GLEAN_10005366 | TTTACCCTCGGCCGATGGAAG/ TGTGAACCATTTGTATCTGGA | 134 | 80.8 | 96.15 | 0.9976 | 89.90 | 0.9966 |

Zhang et al. (2018), *PeerJ*, DOI 10.7717/peerj.5642

**Table 1** (*continued*)

| Gene name | Description | NCBI Homolog locus | Gourdbase Homolog locus | Forward primer sequence/reserve primer sequenc (5′–3′) | Product size (bp) | Tm (°C) | Leaf | | Fruit | |
|---|---|---|---|---|---|---|---|---|---|---|
| | | | | | | | *E* (%) | $R^2$ | *E* (%) | $R^2$ |
| LsCYP[a] | peptidyl-prolyl cis-trans iso-merase 1-like | LOC101206458 | BG_GLEAN_10006142 | CACACCGGCCCTGGTATTTT/ CATCCATGCC TTCAACGACT | 139 | 83.5 | 107.09 | 0.9914 | 95.00 | 0.9941 |
| LsL23A | 60S ribosomal protein L23a | LOC101220073 | BG_GLEAN_10025920 | AAGGATGCCGTGAAGAAGATGT/ GCATCGTAGTCAGGAGTCAACC | 110 | 82.2 | 93.99 | 0.9997 | 98.49 | 0.9958 |
| LsIPT | adenylate isopen-tenyltrans-ferase (cytokinin synthase) | LOC101204427 | BG_GLEAN_10016404 | GCACTCCAATGGCTCGTTTA/ GGTCGATGGTGGATTTGTCG | 89 | 83.0 | 107.39 | 0.9972 | 93.23 | 0.9951 |
| LsDdRP | DNA-directed RNA polymerase subunit | LOC101215872 | BG_GLEAN_10015299 | AAACTCCCTTTCAGCCTCGA/ AGATGTGGCCCTGTTGAGAA | 174 | 81.6 | 95.45 | 0.9984 | 96.07 | 0.9971 |

**Notes.**

Bottle gourd gene ID in the NCBI Database (https://www.ncbi.nlm.nih.gov/) and GourdBase (http://www.gourdbase.cn/) were listed. The two genes labeled as aqua green were selected from RNA-seq data which met the criteria (RPKM >40, RPKMmax/min <2.0) to be candidate RGs for both leaves and fruits. The fourteen genes labeled as light gray were selected from the traditional RG used in Cucurbitaceae crops.

[a] indicated the candidate reference genes selected for following analysis in bottle gourd leaves.

[b] indicated the candidate reference genes selected for following analysis in bottle gourd fruits.

Zhang et al. (2018), *PeerJ*, DOI 10.7717/peerj.5642

**Table 2   Primer sequences and PCR amplification characteristics for candidate reference genes selected from bottle gourd RNA-seq database.**

| Gene name | Description | NCBI Homolog locus | Gourdbase Homolog locus | Forward primer sequence/reserve primer sequenc (5′–3′) | Product size (bp) | Tm (°C) | E(%) | $R^2$ |
|---|---|---|---|---|---|---|---|---|
| **Primers and amplicon characteristics for candidate internal control genes from bottle gourd leaves** | | | | | | | | |
| LsARL | ADP-ribosylation factor-like | LOC103504673 | BG_GLEAN_10014274 | GCTGGTCGAAAGTTGACTCC/GTCAAGGCCAAAGAGTAGGCA | 109 | 82.7 | 106.80 | 0.9988 |
| LsTPT | triose phosphate/phosphate translocator | LOC103483570 | BG_GLEAN_10024475 | ACCACCTACGATTGGCAGAAG/GTCTGGGAAAAGTGGCGGTAT | 140 | 81.7 | 102.65 | 0.9838 |
| LsSRK2I | serine/threonine-protein kinase SRK2I | LOC101206398 | BG_GLEAN_10004502 | TTGACCACTACCCATCTTGCA/GCGAGCCTCATCCTCACTAA | 102 | 81.7 | 90.54 | 0.9945 |
| LsCNX | calnexin homolog | LOC101207554 | BG_GLEAN_10001815 | TCGCTCTCTCATCCCAATCC/GTGCGCATTCTCATTGATGGG | 139 | 86.5 | 101.26 | 0.9940 |
| LsPDI | protein disulfide-isomerase-like | LOC103504071 | BG_GLEAN_10010057 | AGGCCCACTTTGCTTCTTCAA/GAGCAGTCATGACCCTCCAAT | 199 | 82.2 | 98.27 | 0.9967 |
| LsSK | shaggy-related protein kinase | LOC103490499 | BG_GLEAN_10005566 | CTTGCTTCACGTCTGCTTCAA/GTTGTTAGGGAGGCGGACATT | 114 | 81.7 | 95.34 | 0.9935 |
| LsUBC | ubiquitin C | LOC111803940 | BG_GLEAN_10016554 | CACTTGGTGCTTCGTCTCAG/TCGATCGTGTCAGAGCTCTC | 98 | 81.7 | 98.75 | 0.9957 |
| LsGAD | glutamate decarboxylase | LOC103501361 | BG_GLEAN_10014712 | TGTCATAGGGCTTGCCTTCAG/CATTGGGTGATGCTGAGACG | 129 | 84.6 | 99.77 | 0.9966 |
| LsRNC | ribonuclease III domain-containing protein RNC1 | LOC111803262 | BG_GLEAN_10017046 | TACATCTTCAAGTTGCCTGCGT/CCAGAAGTGTACCGGGTTCT | 93 | 81.0 | 94.31 | 0.9976 |
| **Primers and amplicon characteristics for candidate internal control genes from bottle gourd fruits** | | | | | | | | |
| Ls ARIA | arm repeat protein interacting ABF2 | LOC103483725 | BG_GLEAN_10016382 | CTCCCCAATGCAAAAGCTGAC/GAGGTGCTGTTCGACCCTTAA | 82 | 83.9 | 102.99 | 0.9869 |
| LsP4H | prolyl 4-hydroxylase | LOC103485167 | BG_GLEAN_10016694 | AGAGAGAGAGAGGCCTTGGA/CCTGTGTTTC GCCATGGAAAC | 126 | 82.0 | 91.04 | 0.9951 |
| LsXRN1 | 3′–5′exoribonuclease 1 | LOC101214656 | BG_GLEAN_10010883 | ACCTTCCAGATCACACCAGG/AGGCCTCACAGTTCCTCTTC | 129 | 81.2 | 111.01 | 0.9928 |
| LsPARP | inactive poly [ADP-ribose] polymerase | LOC103503572 | BG_GLEAN_10013282 | TTGGAGTCTTCAGGGAGCTG/TCCTCTTGAACGTGGGGTAC | 143 | 81.3 | 93.05 | 0.9970 |
| LsYpgQ | uncharacterized protein YpgQ | LOC103500350 | BG_GLEAN_10019270 | ATGGCGAAAAGAGAAACGGTG/GAAGGATCATGTGACGCGTC | 83 | 84.1 | 115.47 | 0.9986 |
| LsEIF5 | eukaryotic translation initiation factor 5-like | LOC103488416 | BG_GLEAN_10010896 | GCAGCCAATAGTCTCAGCAC/GTAGTTCAAAGTGGAGGGCGT | 142 | 81.6 | 94.77 | 0.9939 |
| LsVAMP | vesicle-associated membrane protein 72 | LOC103502784 | BG_GLEAN_10013195 | AACCTTCGATCTCAGGCACAA/CGCCGCAGACAGACAAAATGA | 145 | 84.4 | 104.32 | 0.9947 |
| LsPL | Phospholipase-like | LOC101210853 | BG_GLEAN_10006169 | CGAATGGGACTCTGCTTTGG/TATTCCGACGAAATCCATCCG | 131 | 83.1 | 96.42 | 0.9925 |
| LsISCA | iron-sulfur assembly protein IscA-like 1 | LOC103489389 | BG_GLEAN_10023226 | ATGGCAGCTTCTTCGTCTTCC/TGGCGCTGTTGAAGAAGTTGT | 127 | 82.7 | 97.43 | 0.9929 |
| LsclpC | ATP-dependent Clp protease ATP-binding subunit ClpC | LOC101207209 | BG_GLEAN_10001965 | TGTGGATGTTGATTCTGATGGA/ACAGGTTACACAGGAATAGCATC | 90 | 79.2 | 94.59 | 0.9971 |
| LsCRCK3 | calmodulin-binding receptor-like cytoplasmic kinase 3 | LOC103487406 | BG_GLEAN_10008798 | ACCGACTGTCCCTTTCACTTG/GTGGCGGATTTTGGATTTGCAA | 83 | 85.0 | 93.72 | 0.9986 |

**Notes.**

Bottle gourd gene ID in the NCBI Database (https://www.ncbi.nlm.nih.gov/) and GourdBase (http://www.gourdbase.cn/) were listed.

## RESULTS

### Transcriptome analysis of *Lagenaria siceraria* under CGMMV infection based on RNA-seq

Bottle gourd leaves and fruits infected by CGMMV were collected from three replicate virus-inoculated plants, and the presence of CGMMV in each sample was further confirmed by RT-PCR and western blot, and the contamination of ZYMV and WMV was excluded by RT-PCR (Fig. S1). The control leaves and fruits samples were in parallel collected from three mock bottle gourds. The analysis of bottle gourd transcripts before and after CGMMV infection showed 639 and 3,930 non-differentially expressed genes (| log2 fold_change | <1, $P \leq 0.05$) in the leaves and fruits of bottle gourd, respectively (Tables S1 and S2). And these non-differentially expressed genes were used as the source of candidate RGs from the RNA-Seq dataset.

### Selection of candidate RGs

In the present study, 11 and 86 candidate RGs respectively from bottle gourd leaves and fruits were screened from our RNA-Seq dataset by setting up a series of conditions (Tables S3 and S4). Only two genes, *LsH3* and *LsWD* (Table S5), met the criteria to be candidate RGs for both leaves and fruits. The primers were designed based on the gene sequences in the database. To select candidate RGs that could be used in both bottle gourd leaves and fruits, these two novel genes with other 14 traditional candidate RGs were used to compare their expression stability. To select candidate RGs that could be used in bottle gourd leaves or fruits, separately, the commonly used reference gene sequences were then compared with the bottle gourd transcriptome data (Table S6). All the 11 candidate RGs of bottle gourd leaves screened from our RNA-Seq data, and seven traditional candidate RGs, *LsPP2A*, *LsADP*, *LsEF1α*, *LsCYP*, *LsRPS15*, *LsTBP*, and *LsRPL23* screened from transcriptome comparison data were selected as candidate RGs of bottle gourd leaves. Of the 86 genes screened from the bottle gourd fruit transcriptome data, the first 11 (based on the ratio of RPKMmax/min) were selected for further analysis in addition to *LsH3* and *LsWD*, and five traditional candidate RGs, *LsPP2A*, *LsADP*, *LsTBP*, *LsTUA*, and *LsGAPDH* screened from transcriptome comparison data were selected as candidate RGs of bottle gourd fruits. Therefore, a total of 16 common RGs of both bottle gourd leaves and fruits (Table 1) and 18 RGs of bottle gourd leaves and fruits separately were screened and analyzed (Tables 1 and 2).

### Evaluation of target specificity and amplification efficiency in RT-qPCR reactions

Preliminary evaluation of candidate reference gene primers was performed by evaluating primer specificity and efficiency. The single peak in melting curve analyses following RT-qPCR confirmed the specific amplification of each gene (Fig. S2). Each amplicon was detected by agarose gel electrophoresis, only a single fragment of the expected size (80–200 bp) was observed (Fig. S3). Further sequencing results showed all genes sequences were exactly 100% identical to those of the corresponding genes in bottle gourd transcriptome databases. Amplification efficiencies of bottle gourd leaves ranged from 90.1% to 110.1%,

whereas those of the fruits ranged from 89.9% to 116.2% (Tables 1 and 2). Furthermore, the standard curves showed good linear relationships (>0.981) between the Cq values and the log-transformed copy numbers of all candidated RGs (Tables 1 and 2). There was no band detected in the negative controls, indicating that the genomic DNA contamination does not exist (Fig. S3).

## Expression intensity of candidate RGs

In order to fully understand the relative expression intensity of all candidate RGs in bottle gourd, three biological and three technical replicates ($n = 9$ for each gene) were used to determine the Cq values for all RGs. From the graph for bottle gourd leaves, the mean Cq values of these candidate genes we selected ranged from 16.51 (*LsCYP*) to 25.63 (*LsTBP*) (Fig. 1A), which represented the highest and lowest accumulation levels, respectively. The minimal variation in gene expression was *LsUBC* (<0.81 cycles) in bottle gourd leaves. The lowest and highest median Cq value of the mRNA accumulation levels in the fruits of bottle gourd was 20.94 (*LsH3*) and 28.88 (*LsTBP*), respectively (Fig. 1B). *LsPLA* expression exhibited the least amount of variation (<1.14 cycles) in bottle gourd fruits. The Cq value 15.18 of *LsCYP* in leaves was the lowest and 28.04 of *LsPP2A* in fruits was the highest (Fig. 1C). The minimal variation in gene expression observed in both leaves and fruits were 2.95 cycles (*LsACT*), 3.06 cycles (*LsADP*) and 3.78 cycles (*LsWD*). However, only the comparison of raw Cq values is not sufficient to evaluate the expression stability of candidate RGs, a further intensive statistical analysis was required for more accurate assessment.

## Comparison of the expression stability of the universal traditional candidate RGs and novel candidate RGs in both bottle gourd leaves and fruits

To select universal candidate RGs expressed stably both in bottle gourd leaves and fruits infected by CGMMV, we screened from our RNA-Seq dataset by setting up a series of conditions (RPKMmax/min < 2, and CV < 0.3, $p < 0.05$, RPKM min > 40). Only two genes, *LsH3* and *LsWD*, met the criteria to be candidate RGs and we then compared the expression stability of these two novel genes with 14 other commonly-used cucurbit RGs. The expression profiles (Fig. 2A) and the variations of the 16 genes (Fig. 2B; Tables S5 and S6) in bottle gourd suggested that *LsWD*, *LsH3* and *LsTBP* and *LsGAPDH* were expressed more stably (CV < 0.3), and the variation in respective expression levels was the lowest among all the genes during CGMMV infection, indicating that these genes, especially the two novel genes *LsWD* and *LsH3*, may be more suitable for normalization than other traditional candidate RGs. In addition, *LsWD*, *LsH3* and *LsGAPDH* each had only one transcript, which could facilitate primer design and ensure the accuracy and reliability of the RT-qPCR compared to other genes.

The stability of these 16 potential RGs was further evaluated with two statistical methods. By geNorm analysis, the average gene expression stability (M) of all of the universal candidate RGs were compared, among them, *LsH3* and *LsGAPDH* showed the lowest M value ($M = 0.284$) in both leaves and fruits, followed by *LsWD* ($M = 0.340$) (Fig. 3A), indicating that these genes displayed the most stable profiles. Pairwise variation $V_{n/n+1}$ was

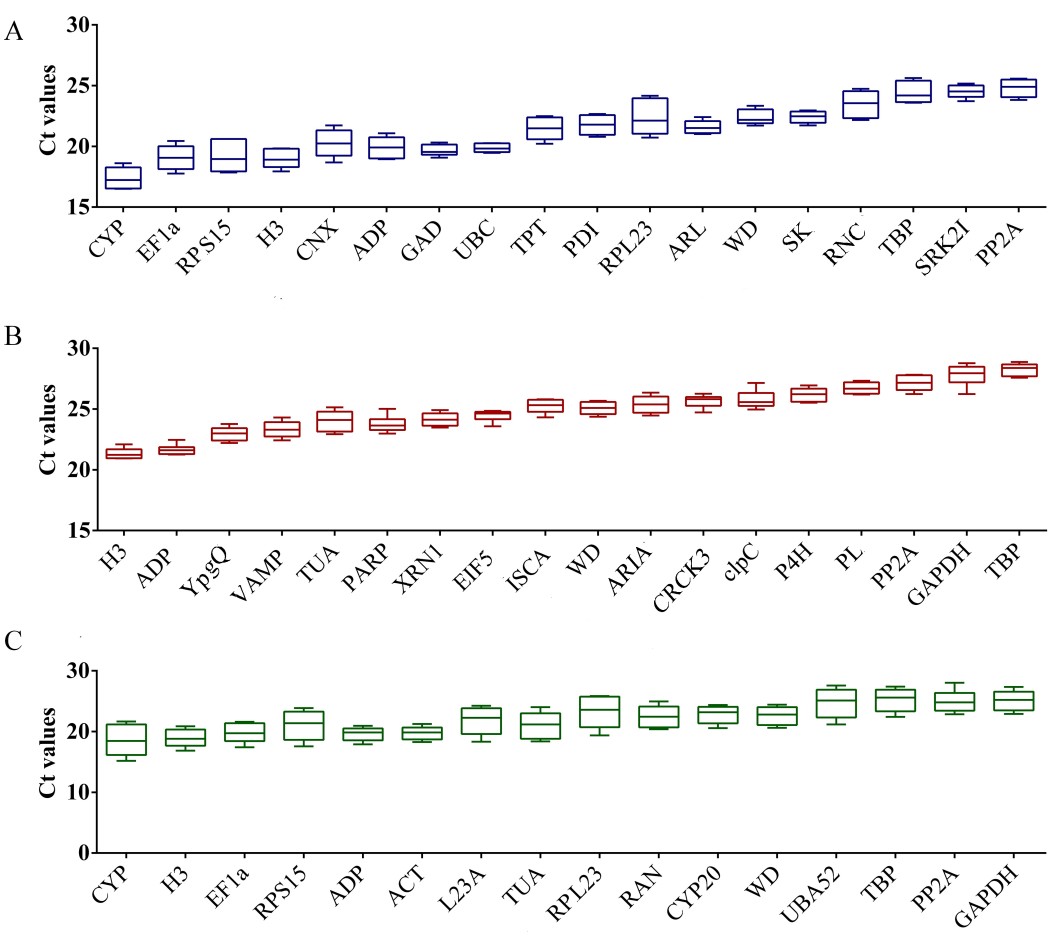

**Figure 1 Expression intensity of candidate RGs in healthy control and CGMMV-infected samples of bottle gourd.** (A) Bottle gourd leaves indicated in blue color; (B) bottle gourd fruits indicated in red color; (C) bottle gourd leaves and fruits indicated in green color. Values are given as Cq (mean of triplicate samples) and are inversely proportional to the amount of template. The box indicates the 25th and 75th percentiles. Whiskers represent the maximum and minimum values. The thin line within the box marks the median.

less than 0.15 in all leaf and fruit samples (Fig. 3B), indicating that adding other RGs was not necessary, and demonstrating that at least two reference genes were required for more reliable normalization, the top two gene were *LsH3* and *LsGAPDH*. The raw Cq values were also transformed into Q values for NormFinder analysis. The lowest stability value of NormFinder analysis indicates the most stably expressed gene. By NormFinder analysis, the best three universal RGs in both leaf and fruit were *LsTBP*, *LsWD* and *LsH3* (Fig. 4A). So, both geNorm and NormFinder analysis suggested that the two novel candidate RGs *LsWD* and *LsH3* were suitable to evaluate the gene expression stability of bottle gourd leaves and fruits infected by CGMMV.
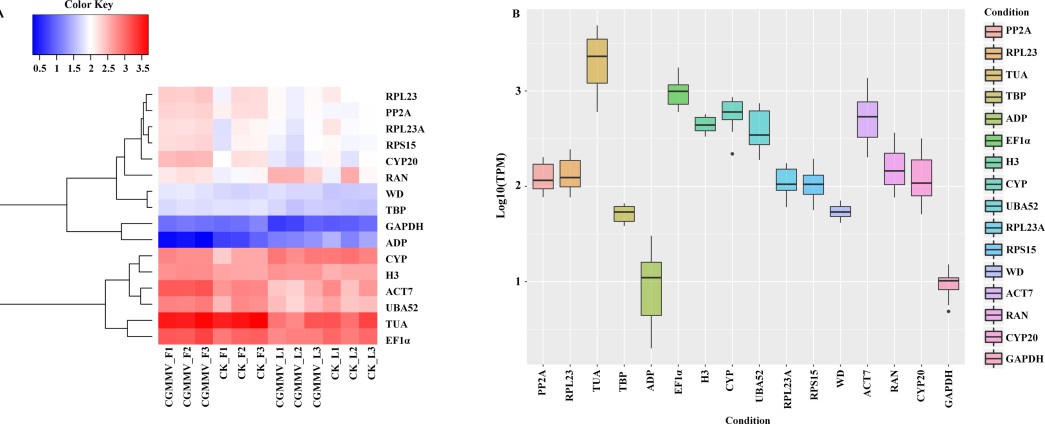

**Figure 2** **Characteristic expression of the universal genes in bottle gourd under CGMMV infection.**
(A) A heatmap was used to visualize the expression pattern of the two novel candidate RGs selected from RNA-Seq data and 14 commonly used RGs under CGMMV infection. (B) Expression levels and variations of the 16 common RGs under CGMMV infection.

## Expression stability of candidate RGs in CGMMV infected leaves and fruits of bottle gourd separately from transcriptome analysis

To further select more suitable candidate RGs expressed stably in bottle gourd leaves or fruits infected by CGMMV separately, 18 RGs of bottle gourd leaves or fruits separately obtained from the RNA-Seq dataset were compared using different algorithms. By geNorm analysis, the average gene expression stability of the 18 candidate RGs of bottle gourd leaves and fruits screened from the RNA-Seq were all less than 1.5, respectively (Figs. 3C and 3E; Table 2). For all the tested leaf samples, *LsTBP* and *LsCYP* showed the lowest M value ($M = 0.213$) (Fig. 3C) of all the candidate RGs while in fruit samples, *LsP4H* and *LsVAMP* had the lowest M value ($M = 0.212$) (Fig. 3E). The pairwise variation $V_{n/n+1}$ of each sample was also less than 0.15 (Figs. 3D and 3F), and the top two reference genes those had the lowest M value were needed for more reliable normalization at least. Similarly, by NormFinder analysis, the best three genes screened from the RNA-Seq dataset were *LsCYP*, *LsH3* and *LsPP2A* in leaves (Fig. 4B), and *LsTBP*, *LsP4H* and *LsXRN1* in fruits (Fig. 4C).

For further clarification, the expression stability of these candidate RGs was examined by two more algorithms. BestKeeper software can only compare the expression levels of up to 10 internal control genes in 100 samples, so only the top ten genes identified by geNorm and NormFinder were selected for subsequent assessment. Of these, the top three candidate internal RGs were *LsCYP* ($r = 0.995$, p-value $= 0.001$), *LsH3* ($r = 0.984$, p-value $= 0.001$) and *LsTBP* ($r = 0.964$, p-value $= 0.002$) in leaf samples, and *LsP4H* ($r = 0.978$, p-value $= 0.001$), *Ls VAMP* ($r = 0.975$, p-value $= 0.001$) and *LsTBP* ($r = 0.959$, p-value $= 0.002$) in fruit samples (Table S7). The results of BestKeeper were therefore broadly consistent with geNorm and NormFinder. We also compared and ranked the tested candidate RGs based on a web-based comprehensive analysis tool, RefFinder, which suggested that the top three candidate RGs screened from the RNA-Seq dataset in bottle gourd leaves were

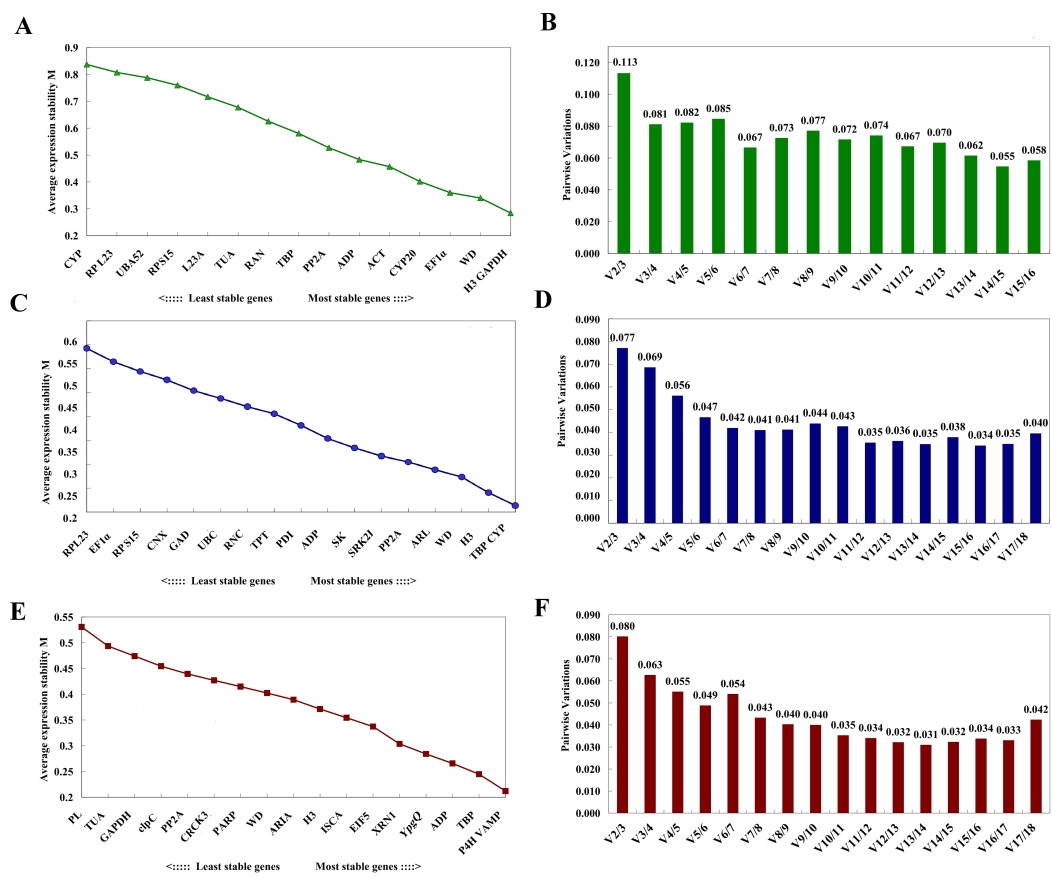

**Figure 3  Expression stability of the candidate RGs analyzed by geNorm.** M represents the stability value. M of RGs screened from RNA-Seq in both bottle gourd leaves and fruits (A), and Vn/Vn+1 of the universal RGs in both bottle gourd leaves and fruits (B); M of RGs screened from RNA-Seq in bottle gourd leaves (C), and Vn/Vn+1 of RGs screened from the RNA-Seq in bottle gourd leaves (D), M of RGs screened from RNA-Seq in bottle gourd fruits (E), and Vn/Vn+1 of the universal RGs in bottle gourd fruits (F).

*LsCYP*, *LsH3* and *LsTBP*, while those in bottle gourd fruits were *LsP4H*, *LsADP*, and *LsTBP* (Table S7). These should therefore be the best RGs to use in RT-qPCR.

## Validation of the candidate RGs

According to the transcriptional data, expression of the *IPT* and *DdRP* genes of *L. siceraria* changed significantly in response to CGMMV. *LsIPT* increased 1.78 fold in leaves and decreased 1.2 fold in fruits compared with their mock-inoculated tissues, while *LsDdRP* increased 1.4 fold in leaves and increased 1.63 fold in fruits (Table S8). These genes were therefore chosen to evaluate the reliability of the top candidate RGs as indicated by the previous analysis. The top rank RGs *LsH3*, *LsGAPDH*, *LsWD*, *LsCYP*, *LsTBP* and *LsP4H* selected by geNorm and RefFinder were used as candidate RGs. Among them, *LsTBP*, *LsWD*, *LsH3*, *LsGAPDH* and *LsCYP* were selected to use as RGs in leaves, and all these genes together with *LsP4H* for fruit.
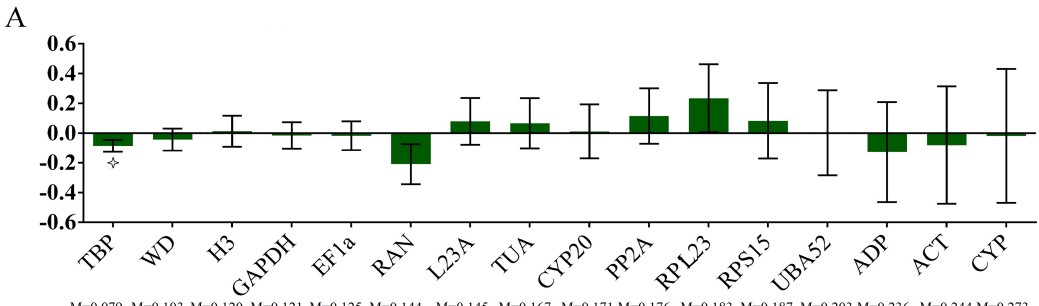

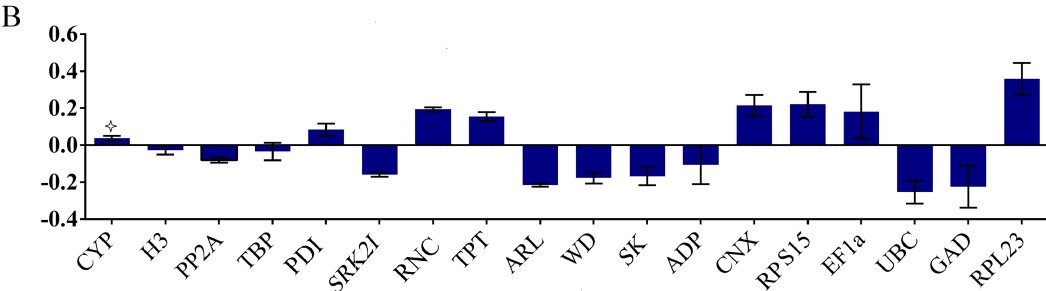

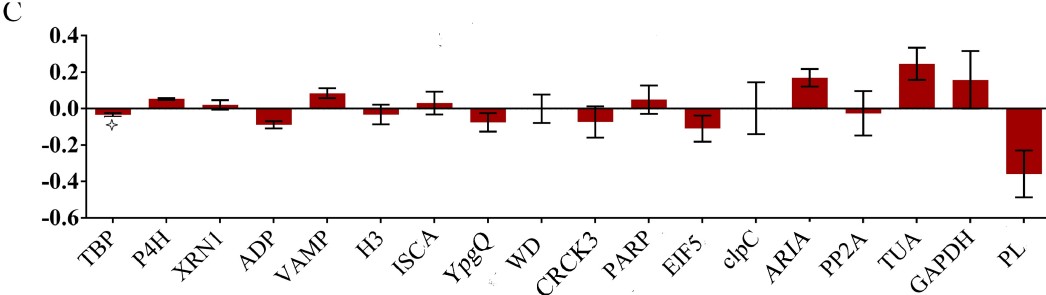

**Figure 4** **Expression stability of the candidate RGs analyzed by NormFinder.** The samples were divided into two subgroups according to the method of leaf and fruit set. The histogram displays the intergroup variation. The error bars represent the intragroup variation. M represents the stability value. Asterisks indicate the best genes. NormFinder analysis of candidate RGs screened from the RNA-Seq and the traditional RGs in both bottle gourd leaves and fruits (A), from the RNA-Seq in bottle gourd leaves (B), and in bottle gourd fruits (C).

*LsIPT* increased 2.75 fold when *LsWD* was used as the reference gene in leaves, with 2.95, 3.82, 3.93 and 4.43 fold increases using *LsGAPDH*, *LsH3*, *LsCYP* and *LsTBP* respectively. The values of *LsIPT* normalized fold expression in fruits were 0.54 (*LsTBP*), 0.52 (*LsWD*), 0.60 (*LsH3*), 0.57 (*LsGAPDH*), 0.59 (*LsCYP*) and 0.56 (*LsP4H*) (Fig. 5). *LsDdRP* increased 1.78, 1.95, 2.51, 2.57 and 2.88 fold in leaves when *LsWD*, *LsGAPDH*, *LsH3*, *LsCYP* and *LsTBP* were the internal RGs respectively, while the corresponding values in fruits were 1.42 (*LsWD*), 1.53 (*LsTBP*), 1.60 (*LsH3*), 1.62 (*LsGAPDH*), 1.80 (*LsP4H*) and 2.31 (*LsCYP*) (Fig. 5). The RT-qPCR results showed that all candidate RGs gave results consistent with the data from the transcriptome database. Overall, the most suitable internal RGs chosen for

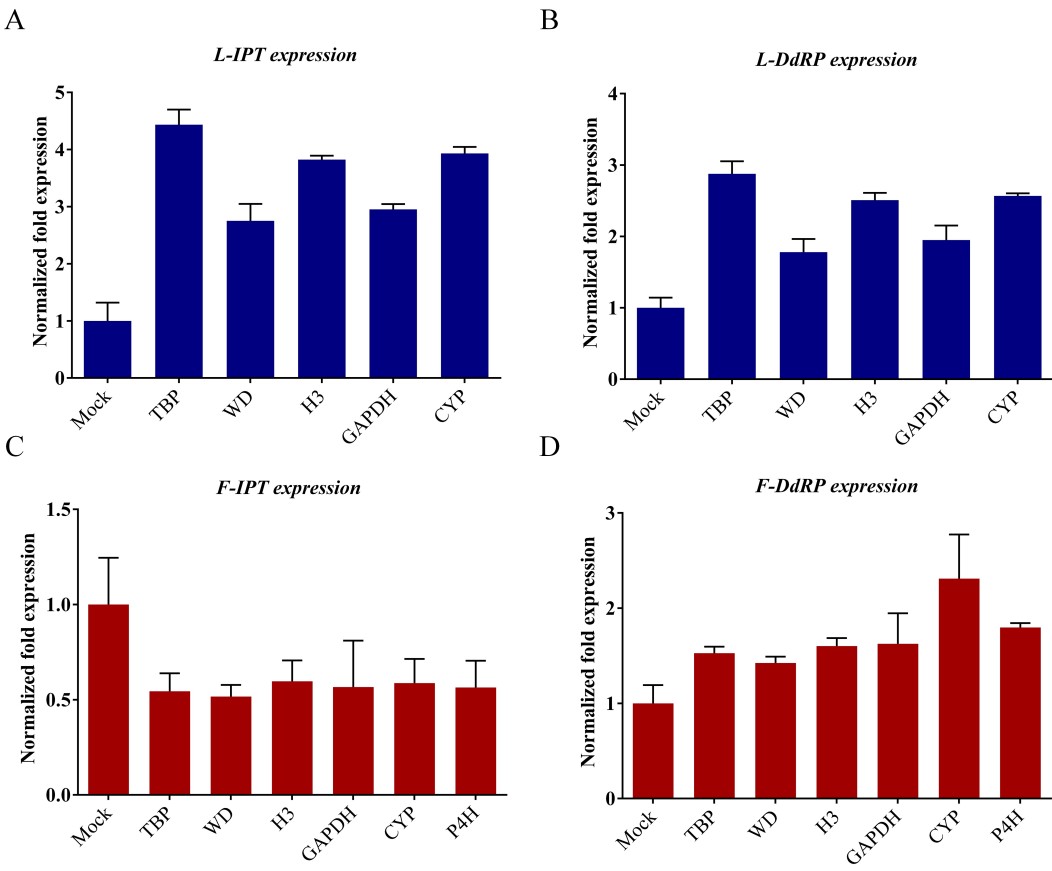

**Figure 5** **Relative quantification of *LsIPT* and *LsDdRP* expression in gourd leaf and fruit infected by CGMMV with RGs selected.** *LsTBP* (leaf and fruit), *LsWD* (leaf and fruit), *LsH3* (leaf and fruit), *Ls-GAPDH* (leaf and fruit), *LsCYP* (leaf and fruit) and *LsP4H* (fruit only) were used as RGs. Error bars represent mean standard error calculated from three biological replicates. A control mock-inoculated sample was used as the calibrator (= 1). (A) *LsIPT* expression of gourd leaves with selected RGs. (B) *LsDdRP* expression of gourd leaves with selected RGs. (C) *LsIPT* expression of gourd fruits with selected RGs. (D) *Ls-DdRP* expression of gourd fruits with selected RGs. L, leaf; F, fruit.

use in leaves were *LsWD*, *LsGAPDH* and *LsH3* and those for fruits were *LsH3*, *LsGAPDH*, *LsP4H* and *LsCYP*.

## DISCUSSION

RT-qPCR, as one of the most commonly used and important tools for gene expression analysis, is characterized by rapidity and efficiency, responsiveness, simplicity in operation, high throughput, and specificity (*Huggett et al., 2005*; *Liu et al., 2012*). RT-qPCR can be used for qualitative or quantitative analysis of gene expression differences, and the appropriate internal RGs for relative quantitative analysis of the expression of genes are essential. Recent studies indicate that there is no single internal reference gene that is absolutely stable and therefore the choice of internal reference gene depends upon the various experimental conditions (*Radonić et al., 2014*; *Ceelen, De Craene & De Spiegelaere,*

2014; *Kong et al., 2014a*; *Kong et al., 2014b*). The ideal internal reference gene should be stably expressed under the corresponding experimental conditions, and its expression level should not be too high or too low; moreover, it should not be a pseudogene and its expression level should not be associated with the cell cycle. With continuous improvement in RT-qPCR requirements, researchers choose two or more RGs for gene analysis in order to reduce the error and obtain more reliable results (*Liu et al., 2012*; *Kong et al., 2016*).

In recent years, transcriptome sequencing technology has been widely used in various fields of molecular biology. The rapid development of transcriptome technology provides a better understanding of gene expression in plant samples from specific tissue, at different developmental stages, or under stress conditions. Transcriptome analysis based on high throughput sequencing can help us to quickly understand the differences in gene expression levels in plant tissue under specific conditions and can also be used to analyze the expression abundance of transcripts, identify the variable splicing of genes, determine the location of transcription, investigate gene fusion events, and discover new transcripts and other important information. In the screening of plant internal RGs, transcriptome sequencing analysis also provides us with a new screening pathway (*Huggett et al., 2005*).

We performed RNA-Seq for CGMMV-infected bottle gourd, and the genes that were not differentially expressed were selected as the candidate RGs based on the transcriptome data. Certain parameters were set to screen the stable expression genes as candidate RGs from RNA-seq data, and some traditional RGs were also compared to identify the most suitable candidate RGs for leaves or fruits, separately. Among them, *LsH3* and *LsWD* were selected from the RNA-seq as candidate RGs in the CGMMV-infected leaves and fruits of bottle gourd. *H3* is one of the most important constituents of chromatin, and its amino acid sequence is highly conserved. Methylation and acetylation of Histone *H3* play an important role in the growth and development of plants (*Bortoluzzi et al., 2017*; *Wollmann et al., 2017*; *Ingouff et al., 2010*). Tryptophan and aspartic acid (WD)-repeat protein is a class of proteins that contain multiple highly conserved WD motifs and are strongly conserved. It is the Gb subunit of heterotrimeric G proteins, which forms a tight dimer (Gbg) with Gg subunits and plays an important role in signal transduction, protein transport, and RNA processing (*Smith et al., 1999*; *Li et al., 2014*; *Van Nocker & Ludwig, 2003*; *Gachomo et al., 2014*). Both *LsH3* and *LsWD* genes were then compared with 14 genes used traditionally in cucurbitaceous crops to select the most suitable RGs in different bottle gourd tissues under CGMMV infection. Both geNorm and NormFinder analysis suggested that the two novel genes *LsWD* and *LsH3* selected from our RNA-Seq data are suitable candidates to use in evaluating the gene expression stability in bottle gourd leaves and fruits infected by CGMMV. The further RefFinder analysis suggested that *LsWD*, *LsGAPDH* and *LsH3* were the best three common optimal RGs for both leaves and fruits whether infected by CGMMV or not. Of the commonly used traditional RGs, *LsGAPDH* was the most stable in both leaves and fruits under CGMMV infection, but the novel *LsWD* reference gene ranked in first place.

Several other novel RGs selected from the RNA-seq data and some traditional RGs were also compared to identify the most suitable candidate RGs for leaves or fruits, separately. geNorm and NormFinder analysis, and the BestKeeper analysis based on these
two algorithms, were consistent with each other with a slight difference, and a web-based comprehensive analysis tool RefFinder combined these analyses and suggested that the top three candidate RGs screened from bottle gourd leaves were *LsCYP*, *LsH3* and *LsTBP*, while those in fruits were *LsP4H*, *LsADP*, and *LsTBP*. These should be the best RGs to use in RT-qPCR.

IPT is an important rate-limiting enzyme in the synthesis of cytokinin (CTK), catalyzing the decomposition of isopentenyl pyrophosphate and adenosine monophosphate to produce isoforms as precursors of CTK (i.e., monopentenyl AMP, iAMP), which can promote the increase in CTK content in plant cells (*Hwang & Sakakibara, 2006*; *Zhu et al., 2012*). The expression of the *IPT* gene in plants can improve stress resistance (*Reguera et al., 2013*; *Žižková et al., 2015*), delay leaf senescence, and improve defence against insect pests (*Smigocki et al., 1993*; *Novák et al., 2013*). DdRP is an essential enzyme for the replication of transcription systems in a variety of organisms and plays an important role in controlling transcription during gene expression (*Wnendt et al., 1990*; *Knopf, 1998*). These two genes were selected to validate the applicability of the screened RGs from bottle gourd in response to CGMMV. According to the comprehensive analysis, *LsTBP*, *LsWD*, *LsH3*, *LsGAPDH* and *LsCYP* were selected to as RGs in leaf, and all these genes with *LsP4H* were used to analyze *IPT* and *DdRP* expression in fruit. RT-qPCR results combined with transcriptome analysis showed a consistent trend of expression, which indicated that the candidate RGs were stable. Among these genes, *LsWD*, *LsGAPDH* and *LsH3* were most suitable as internal RGs in the leaf, and *LsH3*, *LsGAPDH*, *LsP4H* and *LsCYP* as those for the fruit. Therefore, the novel genes *LsH3* and *LsWD* were more stable both in leaves and in fruits under CGMMV infection than the previous reference genes, such as *CYP*, *GAPDH*, and *TBP*, although among the traditional RGs, *GAPDH* showed its superiority both in leaves and in fruits under CGMMV infection.

For the limitation that the RNA-Seq data was only from one bottle gourd variety infected with CGMMV, we further analyzed the existence of these RGs in different bottle gourd genotypes and found all these RGs are existed in about 50 different type of bottle gourd with different fruit shape according to the resequencing data we could access. Also, in the single nucleotide polymorphism (SNP) analysis about the sequences of these RGs amplified with the primers we designed, most are very conserved with no variation, only four RGs (*LsXRN1, LsUBC, LsCYP, LsTUA*) had slight variation. The SNP analysis further suggest the conservation of these RGs in different bottle gourd type. These selected RGs for bottle gourd leaves and fruits lay the foundation for further related research.

## CONCLUSIONS

In this study, 16 candidate RGs of both leaf and fruit and 18 candidate RGs mostly from separate RNA-Seq datasets of bottle gourd leaf or fruit were assessed for their potential use as RGs in bottle gourd. Reliable normalized analysis by geNorm, NormFinder, BestKeeper and RefFinder indicated that *LsWD*, *LsGAPDH* and *LsH3* were the most optimal RGs for bottle gourd leaves, and *LsH3*, *LsGAPDH*, *LsP4H* and *LsCYP* for the fruit. The candidate

RGs provided in this study could be used to normalize the target genes in bottle gourd leaves and fruits to improve the accuracy and reliability of gene expression studies and the further related studies.

## ACKNOWLEDGEMENTS

We would like to thank Prof. M. J. Adams, Rothamsted Research, Harpenden, Herts, UK for correcting the English of the manuscript.

### Funding

This work was financially supported by Special Fund for Agro-Scientific Research in the Public Interest (201303028), Science and Technology Program (Public Service Technology Application) of Zhejiang Province (2017C32040), the National Nature Science Foundation of China (31500124) and Science and technology fund of Putian City (2016S3001). The funders had no role in study design, data collection and analysis, decision to publish, or preparation of the manuscript.

### Grant Disclosures

The following grant information was disclosed by the authors:
Agro-Scientific Research in the Public Interest: 201303028.
Science and Technology Program (Public Service Technology Application) of Zhejiang Province: 2017C32040.
National Nature Science Foundation of China: 31500124.
Science and technology fund of Putian City: 2016S3001.

### Competing Interests

The authors declare there are no competing interests.

### Author Contributions

- Chenhua Zhang performed the experiments, analyzed the data, prepared figures and/or tables, authored or reviewed drafts of the paper, approved the final draft.
- Hongying Zheng conceived and designed the experiments, performed the experiments, prepared figures and/or tables, authored or reviewed drafts of the paper, approved the final draft.
- Xinyang Wu and Xiaohua Wu performed the experiments, approved the final draft.
- Heng Xu, Jiejun Peng and Guojing Li analyzed the data, approved the final draft.
- Kelei Han, Yuwen Lu and Lin Lin contributed reagents/materials/analysis tools, approved the final draft.
- Pei Xu approved the final draft.
- Jianping Chen authored or reviewed drafts of the paper, approved the final draft.
- Fei Yan conceived and designed the experiments, prepared figures and/or tables, authored or reviewed drafts of the paper, approved the final draft.
## Data Availability

The raw data are provided in the Supplemental Files.

## Supplemental Information

Supplemental information for this article can be found online at http://dx.doi.org/10.7717/peerj.5642#supplemental-information.

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
