# Peer review of "Genome-wide identification of new reference genes for RT-qPCR normalization in CGMMV-infected Lagenaria siceraria"

_PeerJ, doi:10.7717/peerj.5642_

## Round 0.1 · original submission · Minor Revisions

Dear Dr Chenhua,

Thank you for your interest in Peer J. I have received the contributions of the independent reviewers. All the reviewers were very positive about the work described, but there it is my opinion that the MS deserves minor revisions, particularly in what concerns the MIQE guidelines and the discussion.

Looking forward to receive the revised MS,

Sincerely
Ana I Ribeiro-Barros

Reviewer 1 ·

Basic reporting

Authors use clear and professional language throughout the paper.
The structure is suitable as well as the figures and other information available.

Experimental design

Relevant work to ensure the robustness and accuracy of future gene expression studies

Methods described with sufficient detail and information.
Anyway, some consideration should be taken into account:

Line 121 to 127 -once RNA extraction is described these paragraphs should be reformulated and included in the point: “RNA and first strand cDNA preparation”

Line 129- “RNA sequencing”: although the authors have used the RNA-seq data for the selection of candidate genes, this point is beyond the scope and objectives of this article. Should be withdrawn.

Line 177 – 10mM/ml should be replaced by 10mM

Validity of the findings

Line 284 to 286- “…stability of the 18 and 18 candidate…” should be replaced by:”…stability of the 18 candidate….”

Line 429- to 431 (legend Fig 2)- “As shown in the figure [….] infection” this is a result/conclusion, therefore should be included in the text and not in the legend.

Additional comments

No comment

Reviewer 2 ·

Basic reporting

The manuscript is well-written in an unambiguous and technically correct style. The Authours used a professional English language.
Introduction is well-structured. The background of your study provide context to the information discussed throughout this paper. Although the background information include both important and relevant studies, I suggest to read and add in the introduction this recent reference: Wamg et al. GourdBase: a genome-centered multi-omics database for the bottle gourd (Lagenaria siceraria), an economically important cucurbit crop. Scientific Reportsvolume 8, Article number: 3604 (2018).
The structure of the paper is conform to format of ‘standard sections’.
Figure and tables should be improved. The Authors should decide if reported the same value in graphs or table, in this way are redundant.

In the paper all results are relevant to the hypothesis and are not sufficiently argumented by discussions

Experimental design

This paper matches within the Aims & Scope of the journal. This research is relevant and meaningful to improve the knowledge in the plant-pathogen intercation (Lagenaria-CGMMV).
This is the first study that evaluated the candidate reference genes in this pathosystem.
M&M are well-described and structured, with high technical standard. All methods are reproducible by another investigator. Some aspects should be clarify (see comments of Authors)

Validity of the findings

The paper contains some interesting and useful data, and, thus, yields new information in this research area. Data are robust but there are some problems in the stastistical analysis.
Discussions and conclusions should be improved.

Additional comments

The manuscript “Genome-wide identification of new reference genes for qRT-PCR normalization in CGMMV-infected Lagenaria siceraria” by Zhang et al. represents a comparison and choosing of reference genes. This is the first study in which a set of candidate RGs was analyzed in terms of their expression stability in bottle gourd infected by cucumber green mottle mosaic virus.
The author used three different statistical algorithms to compare these reference genes and obtained the most stable genes. In my opinion, all data are reliable and sufficient. Furthermore, the Authors evaluated the expression analysis of two target genes, chosen for RGs validation, involved in adenylate isopentenyltransferase (IPT) and DNA-direct RNA polymerase subunit (DdRP).
The data obtained and discussed are useful and provide novel insights into this area of applied research.
However, some parts of the manuscript need some improvements before the publication.
In particular:
Title
The authors should change “qRT-PCR” into “RT-qPCR” and check in whole paper.

Abstract
L30: RGs are not preferred, but they are the most stable identified in this study.

Introduction
L40: The Authors should improve this sentence. Lagenaria siceraria is a specie belongs to Cucurbitaceae family.
L43 Change “utilitarian” into “horticultural”
L97: Correct the typing error
L100: RefFinder is an online available tool that integrates some algorithms. On the basis, of your data (table 2 and table 2) we cannot assume 100% efficiency for all genes and RefFinder outputs may be biased because PCR efficiencies are not taken into account. I suggest to delete this analysis.

M&M
The Authors should improve the description of the number of RGs obtained by literature and RNA-Seq and explain this in the text and in the captions of figures and tables.

L121-124: The authors should report these informations in the “RNA and first strand cDNA preparation” section
L124: Add this method “microfluidic capillary electrophoresis with the Agilent 2100 bioanalyzer”. Check the brand of Bioanalyzer.
L127: Add the reference.
L125: The authors should add some information on the extraction of CGMMV by infected plant.
L140-141: The Authors should improve this sentence and delete “could”.
L162: In this section the Authors should add the method used to obtain PCR efficiency (E) for each primer pair and regression coefficient (R) as suggested by the guidelines “Minimum Information for Publicationof Quantitative Real-Time PCR Experiments (MIQE)” (Bustin et al. 2009)
L182: The authors should indicate quantification cycle as “Cq”, according to the RDML (Real-Time PCR Data Markup Language) data standard (http://www.rdml.org), and reported by the guidelines “Minimum Information for Publicationof Quantitative Real-Time PCR Experiments (MIQE)” (Bustin et al. 2009).
L187: Delete RefFinder
L193 and L202: Correct the typing error
L209: The Authors should improve this section, because some RGs are reported in Table 1 and other in table 2.
L217: The name reported for some genes are not in agreement to Table 1
L220: Check the number of RGs, in Table 2 there are 20 RGs.

L227 (Fig.S2): Negative control should be add in each experiment, as recommended by MIQE guidelines. In Fig. S2 the negative control are not indicated in the melting curve.
L229 (Fig. S3): The Authors should improve the text at the bottom of the panel A.
L233: Delete R2, the values of single RG are reported in the tables.
L285: The Authors should improve this sentence
L300-305: Delete these sentences.
L312-314. Combined use of geNorm, NormFinder and BestKeeper to select and validate the best RGs generated substantial discrepancies in the final ranking due to different mathematical models associated with each algorithm, such as demonstrated in previous studies. To overcome differences in the ranking of RGs, how did the Authors obtain a final ranking? Explain the used method in Materials and Methods.
L313-314: The authors should choose the number of RGs on the basis of the results obtained by GeNorm (L271-272; L290)

Discussion
L328-338: Add some references.
L348-374: The discussion should completely rewritten on the basis of correct interpretation of their interesting results.

Conclusion
The conclusion should completely rewritten on the basis of correct interpretation of their interesting results.


Table 1: Check the name of gene and add the reference for each RGs.
Table 2: Check the name of RGs
Figure 3: The Authors should add the means of “M”
Table 3: The Authors reported in this table the values indicated in Fig. 3A and 4A, respectively. In Fig. 4A some values are negative, check these.
Table 4: The Authors reported in this table the values indicated in Fig. 3C,3E and 4B, 4C, respectively. In Fig. 4B and 4C some values are negative, check these.
The Authors should decide if reported the same value in graphs or table, in this way are redundant.

Reference: Check the reference in the manuscript

Reviewer 3 ·

Basic reporting

The manuscript by Zhang et al. brings an important contribution for the genetics and breeding of cucurbitaceae. The species has not only economical potential per se, but also as a rootstock source for watermelon and other cucurbit crops. The use of appropriate reference genes is of paramount importance for the normalization of gene expression and the proper identification of differentially expressed genes in transcriptomic analyses.

Experimental design

The experimental design used by the authors is correct and the tools (software) used to evaluate/validate the reference genes are the current used by the scientific community. The authors were successful in identifying reference genes for leaves ( LsWD, LsGAPDH and Ls H3) and fruits (LsH3, LsGAPDH, Ls P4H and Ls CYP) of Lagenaria siceraria. The use of GeNorm and NormFinder, as well as Bestkeeper and Reffinder was appropriate, allowing the authors to validade and demonstrate the quality of their results. Vn/Vn+1 checks were used to indicate that the group of RG candidates was adequate.

Validity of the findings

The findings are relevant, since they can provide tools for geneticists/breeders working on this crop for a better assessment of genes involved in the response to CGMMV. The correct identification of genes responding to infections can expedite the process of identifying resistance genes and transfer these to commercial genotypes.

Additional comments

Clear, well written manuscript and elegantly presented. The objectives were fulfilled.

Reviewer 4 ·

Basic reporting

I think that this manuscript was informative, systematic, logical and comparatively well written although several typos to be corrected stood out in the manuscript.

Experimental design

Each experiment was generally well organized to achieve the objectives. However, I can’t hardly point out my concerns as below.
1. Do the authors believe that the newly developed RGs are applicable to other bottle gourd genotypes. I would like to hear the authors’ logic explanation on this issue in addition to the genotypic situation (an inbred line, F1 hybrid, local variety with heterozygosity/heterogeneity, and so on) of the accession ‘Hangzhou Gourd.’
2. Why didn’t the authors clarify the race or isolate name of CGMMV used as the inoculum. Similarly with the upper question, why did the authors exclude an unstable and/or altered expression pattern of the new RGs depending on races?

Validity of the findings

Collectively, the authors newly suggested the stably expressed reference genes for qRT-PCR normalization in the bottle gourd plants infected by CGMMV based on the results obtained from various approaches. They might be worthy of notice for the genomic studies of the bottle gourd with a long history. However, it is a flaw that the novel RGs have limited availability, that is, restrictively in CGMMV-infected condition.

Additional comments

3. The authors should show the pictures of leaves and fruits with typical symptoms caused by CGMMV, merged with Fig. S1.
4. What are the methods for cultivation and management of bottle gourd performed according to Li et al. (2016). Li et al. (2016) described those very briefly in their article. Please provide enough details, which set readers at ease from apprehension regarding any biotic/abiotic stresses.
5. Please mention when the fruits sampled, and the total RNAs were extracted from which part of the fruits.
6. I recommend that the authors put forward evidence to support that the bottle gourd plants were not contaminated by other viruses such as WMV and ZYMV, which can easily infect bottle gourd plants as well.

---

## Round 0.2 · accepted · Accept

Dear Dr Chenhua,

It is my pleasure to inform you that the revised version of your MS "Genome-wide identification of new reference genes for RT-qPCR normalization in CGMMV-infected Lagenaria siceraria" is now accepted for publication in Peer J. Thank you for choosing PeerJ.

Best regards

Ana Ribeiro-Barros

# Reviewer 2 ·

Basic reporting

The manuscript is well-written in an unambiguous and technically correct style. The Authours used a professional English language. Introduction is well-structured and a new reference has been added as suggested my previous revision. The structure of the paper is conform to format of ‘standard sections’. Figures and tables have been improved. In the paper all results are relevant to the hypothesis and are sufficiently argumented by discussions

Experimental design

New information have been added

Validity of the findings

The paper contains some interesting and useful data, and, thus, yields new information in this research area. Data are robust and some information in stastistical analysis has been provided.
Discussions and conclusions have been improved.

Additional comments

The Authors reported in the revised manuscript, all suggestions recommended by reviewers. Revised paper has been improved and discussion and conclusion have been rewritten on the basis of their obtained results.